# Meaningful Activities and Psychosomatic Functions in Japanese Older Adults after Driving Cessation

**DOI:** 10.3390/ijerph182413270

**Published:** 2021-12-16

**Authors:** Atsushi Nakamura, Michio Maruta, Hyuma Makizako, Masaaki Miyata, Hironori Miyata, Gwanghee Han, Yuriko Ikeda, Suguru Shimokihara, Keiichiro Tokuda, Takuro Kubozono, Mitsuru Ohishi, Takayuki Tabira

**Affiliations:** 1National Institute for Minamata Disease, Ministry of the Environment, 4058-18 Hama, Minamata, Kumamoto 867-0008, Japan; 2Doctoral Program of Clinical Neuropsychiatry, Graduate School of Health Sciences, Kagoshima University, 8-35-1, Sakuragaoka, Kagoshima 890-8544, Japan; 814.miya.418@gmail.com (H.M.); k5848730@kadai.jp (S.S.); 3Department of Rehabilitation, Okatsu Hospital, 3-95, Masagohonmachi, Kagoshima 890-0067, Japan; m.maru0111@gmail.com; 4Graduate School of Health Sciences, Kagoshima University, 8-35-1, Sakuragaoka, Kagoshima 890-8544, Japan; makizako@health.nop.kagoshima-u.ac.jp (H.M.); miyatam@m3.kufm.kagoshima-u.ac.jp (M.M.); yuriko@health.nop.kagoshima-u.ac.jp (Y.I.); tabitaka@health.nop.kagoshima-u.ac.jp (T.T.); 5Department of Neuropsychiatry, Kumamoto University Hospital, Kumamoto, 1-1-1 Honjo Chuo-ku, Kumamoto 860-8556, Japan; hans11057@gmail.com; 6Department of Rehabilitation, Kirameki Terrace Healthcare Hospital, 43-30 Kouraicho, Kagoshima 890-0051, Japan; gomyway.k.t@icloud.com; 7Department of Cardiovascular Medicine and Hypertension, Graduate School of Medical and Dental Sciences, Kagoshima University, Kagoshima 890-0075, Japan; kubozono@m.kufm.kagoshima-u.ac.jp (T.K.); ohishi@m2.kufm.kagoshima-u.ac.jp (M.O.)

**Keywords:** driving cessation, meaningful activities, community-dwelling older adults, psychosomatic functions

## Abstract

The purpose of this cross-sectional study was to analyse the differences in meaningful activities and psychosomatic function depending on the driving status of community-dwelling older adults. Data from 594 older adults were obtained, including activities meaningful to individuals and psychosomatic functions, such as grip strength, depression, cognitive function, and ability of activity. Participants were divided into active driving (n = 549) and after driving cessation (n = 45) groups. In addition, the active driving group was operationally divided into three groups: high-frequency group (n = 387), medium group (n = 119), and infrequent group (n = 42). In the after driving cessation group, grip strength, and Japan Science and Technology Agency Index of Competence scores were significantly lower. Furthermore, the proportion of apathy and physical and social frailty was significantly higher in the after driving cessation group. Regarding meaningful activity, domestic life scores in the after driving cessation group were significantly higher than those of the active driving group. Decreased driving frequency in the active driving group was associated with weak muscle strength, lack of interest, and low activity. This study demonstrated that meaningful activity differed based on the driving status. Hence, we should support the activities of older adults who are considering driving cessation.

## 1. Introduction

In Japan, the number of driver’s license holders aged 75 years and above is increasing, causing many fatal accidents [1]. To resolve this problem in recent years, a supportive environment that involves, for example, driving aptitude consultations and support measures for older drivers considering driving cessation, has been promoted to make it easier for them to return their driver’s licenses. However, previous studies on driving cessation have reported that musculoskeletal and neurological problems, visual problems, and cognitive decline lead to driving cessation in older adults. Furthermore, problems with the musculoskeletal system and vision [2], being a woman, and decline in independence as measured in activities of daily living (ADL) [3] have been reported as reasons for driving cessation in older adults. In contrast, driving cessation in older adults leads to poor health [4]. This is because it decreases physical function and increases the risk of functional limitation [5,6], cognitive decline [7], depressive symptoms, depression [8,9,10], frailty [11,12], and mortality [13]. Therefore, driving cessation can be caused by diminished physical and psychosomatic function. Further, environmental factors such as financial reasons can also affect driving cessation. In older adults, driving cessation can impair physical and psychosomatic function, and personal factors such as their changing interests. Therefore, considering complex interactions between physical function, activity, participation, environmental and personal factors with respect to driving cessation is necessary. Moreover, it has been reported that driving cessation in older adults negatively impacts life satisfaction, time spent outside [14], networking with friends [15], paid work, and volunteering [16]. Therefore, in addition to creating an environment that makes it easy for older adults to return a license, it is also essential to prevent the deterioration of their physical function and activity that occurs after the suspension of driving.

Tarumizu City (Area: 62.49 square miles), with a population of 14,379 and a low population density, is one of the least populated areas of Japan, and its population aging rate is extremely high. In addition, public transportation is inadequate and driving is necessary to perform various essential life functions. Participation in activities is essential for older adults as it benefits their health and improves their psychological well-being and health-related quality of life [17,18]. In 2002, the World Health Organization developed “Active Aging” to respond to the progress of global aging, and Active Aging places a great deal of emphasis on participation in activities that individuals find meaningful [19]. Quite a few meaningful activities (e.g., cooking meals, dressing, and bathing) are routine, while others include work, caring for others, social activities, and leisure activities [20]. Meaningful activities are those that include personal purposes and values rather than merely indicating the activity [21]. In addition, previous studies have reported a growing recognition for engaging in personally valued activities (meaningful activities) that are also beneficial for the well-being of older adults [22]. Therefore, engaging in meaningful activities is crucial for maintaining good health. Compared to men, women older than 65 years are reported to have a significantly higher self-assessment of their labour in their domestic life [23]. Driving interruptions have been reported to be associated with low levels of out-of-home activities [24] and productive engagement [16]. It has also been reported that, in rural older people, driving is important for the activities they want or need to perform [25]. However, there are no reports regarding the activities that older people, who have stopped driving or drive infrequently, value. Therefore, it is necessary to investigate the content quality, satisfaction, and performance of meaningful activities for older adults who have stopped driving. The reduced frequency of driving may also be associated with reduced activity.

This cross-sectional study aimed to analyse the differences in meaningful activity and mental and physical function between community-dwelling older adults with different driving statuses (driving or not driving, gender, frequency of driving). Understanding how reduced outings and driving affect the physical and mental functioning and activity of older people is useful while considering pre- and post-driving cessation support preparations.

## 2. Materials and Methods

### 2.1. Participants

This cross-sectional study used data from the Tarumizu Study 2018. It was a collaborative study undertaken by Kagoshima University, the Tarumizu City office, and Tarumizu Chuo Hospital. This study was conducted from June to December 2018 as a community-based health survey. Reply-paid postcards were mailed to the residents of Tarumizu City who were aged 40 years or older at the time of examination, and residents were recruited through local newspaper advertisements and community campaigns. The recruitment period was from April to June 2018, and 1385 people participated in the survey. The survey was conducted at public facilities in Tarumizu, and participants attended one of 24 sessions a year.

This study targeted citizens over the age of 65. The exclusion criteria were as follows: Participants whose data regarding the Aid for Decision-making in Occupation Choice (ADOC) (n = 8) and questions about driving (n = 4) were missing; participants who had a history of stroke (n = 36), Parkinson’s disease (n = 1), and dementia (n = 8); and participants who had never owned a driver’s license (n = 208) in their lifetime.

We questioned the participants about their present driving status and then divided them into two groups: the active driving (n = 549) group and the after driving cessation (n = 45) groups. The active driving group included people with a license, and the after driving cessation group included those who had returned their license (n = 27), who had not renewed their license (n = 2), and who had not returned their license, but were not currently driving (n = 16). In addition, the active driving group was operationally divided into three groups: high-frequency group (n = 387, 6–7 days a week), medium group (n = 119, 3–5 days a week), and infrequent group (n = 42, 2 days or less a week), and their characteristics were compared. Finally, data from 594 community-dwelling older adults (age ≥65 years, mean age: 73.5 ± 5.8, women: 53.2%) were analysed for this study (Figure 1). The ethics committee of Faculty of Medicine, Kagoshima University approved the study protocol (approval number 170351).

### 2.2. Meaningful Activity

In this study, meaningful activities were operationally defined as “activities that individuals consider important in their daily life” [21]. The ADOC was developed as a meaningful activity choice for clients in rehabilitation [26]. The ADOC consists of eight categories: self-care, mobility, domestic life, work/education, interpersonal interaction, social life, sport, and leisure. The eight categories contain 95 illustrations related to “Activities and Participation”, including the International Classification of Functioning, Disability, and Health. Leisure is defined as those activities which produce intrinsic rewards and provide the participant with life-enhancing meaning and a sense of pleasure [27]. The leisure activities in ADOC include 29 items, such as painting, reading, making sweets, gardening, and travelling. An English version has already been developed [28], and the validity of satisfaction in the evaluation has been reported [29]. It is also a preferable tool for focusing on meaningful personal activities [30]. The ADOC has a visually significant effect [31] and is an effective tool for eliciting information about meaningful activities from community-dwelling older adults. Data were collected via a face-to-face survey between the researcher and the subjects. The subjects were shown 95 illustrations of ADOC and asked verbally, “What are the meaningful activities in your daily life?” Therefore, participants were asked to select three to five meaningful activities from the ADOC and then rank the selected activities. Participants evaluated satisfaction using a scale of 1–5 for the selected meaningful activities ranked by ADOC (1 = very dissatisfied, 5 = very satisfied). Furthermore, we also used a scale of 1–10 to assess performance with the selected activity (1: with great difficulty, 10: perfectly). Satisfaction and performance are measures of the individual meaningful activity chosen by them. The researchers in this study were occupational therapists and a few occupational therapy students. Before beginning the study, we conducted two lectures, about two hours each, on the investigation method of meaningful activities. In addition, on the day of the study, we conducted approximately 30 min of practical training before the survey. The study is conducted 24 times a year, and the researchers are engaged multiple times.

### 2.3. Psychosomatic Functions

Regarding psychosomatic function, we examined the participants’ depressive state [8,9,10], cognitive function [7], frailty [11,12], and apathy [32], which have been reported to be related to driving cessation in previous studies. Their depressive state was evaluated using the Geriatric Depression Scale (GDS-15), and 5 points or more (out of 15) indicated a depressive state [33]. Moreover, among the subordinate items of GDS-15, questions like (1) “Have you dropped many of your activities and interests? (yes)”, (2) “Do you prefer to stay at home, rather than going out and doing new things? (Yes)”, and (3) “Do you feel full of energy? (No)” were judged as apathy if the answers to these scored ≥2 points [34]. Apathy is a behavioural symptom defined as disinterest and loss of motivation [35]. Cognitive function was assessed using a Mini-Cog consisting of a three-word recall task and a clock drawing test [36]. The total scores were the sum of the correct words recalled (0–3) and the drawing of the clock (0 or 2), with a cutoff of <3, which was reported to distinguish between people with and without cognitive impairment [36,37]. Therefore, in this study, a total score of <3 was defined as poor cognitive function.

Frailty is a state of physical and mental decline due to aging, which involves the interaction of physical, cognitive, and social aspects. Physical frailty was evaluated for five items: weight loss, weakness, exhaustion, slowness, and low levels of activity, with reference to the definition of the Cardiovascular Health Study (CHS) [38] and the report by Makizako et al. [39]. Physical frailty was recognised if three or more of the five items applied to participant; pre-frailty was not included in the study. Cognitive frailty is defined as the presence of both physical frailty and cognitive impairment. Cognitive frailty was assessed using the National Centre for Geriatrics and Gerontology-Functional Assessment Tool (NCGG-FAT) to define disability corresponding to the population base. This aspect included community-dwelling older adults (score ˃1.5, standard deviations (SD) below the age- and education-specific mean). NCGG-FAT consists of four domains: memory, attention, executive function, and processing speed [40]. Those with either decreased slowness (if <1.0 m/s walking speed regardless of gender and height) or weakness (if <26 kg grip strength for men, if <18 kg grip strength for women), and cognitive impairment were considered cognitively frail [41]. Social frailty considered five questions about social rules, daily social activities, and social relationships: living alone (yes), going out less frequently than last year (yes), visiting friends sometimes (no), wanting to help friends or family (no), and talking with someone every day (no) [42]. If two or more of them were acknowledged, participants were considered socially frail, excluding pre-frailty [42]. The Japan Science and Technology Agency Index of Competence (JST-IC) was used to evaluate the activity’s ability. It was developed as an index to evaluate whether older Japanese adults can live independently and greater actively alone [43,44]. It included four areas (16 items): technology usage, information practice, life management, and social engagement. Higher scores reflected higher activity competence (range: 0–16).

### 2.4. Statistical Analysis

We used Student’s *t*-test for continuous variables, Pearson’s χ^2^ tests for categorical variables, and the Mann–Whitney U-test and Kruskal–Wallis test for ordinal variables. To exclude the effect of gender, we performed the same analysis only in women. In addition, the active driving group was operationally divided into three groups: high-frequency, medium, and infrequent groups, and statistical analysis was performed. We used one-way ANOVA for continuous variables, Pearson’s χ^2^ tests for categorical variables, and Kruskal–Wallis test for ordinal variables. Meaningful activities were analysed, including all selected activities from the first to fifth place. Further, the representative values of satisfaction and performance were taken as the median score of first to fifth place. All analyses were conducted using IBM SPSS Statistics 24.0 (IBM Corp., Armonk, NY, USA), and *p* values < 0.05 were considered statistically significant.

## 3. Results

The characteristics of the study participants are listed in Table 1. The 594 participants were divided into two groups: active driving (n = 549) and after driving cessation (n = 45). The age and proportion of women in the after driving cessation group were significantly higher (age: *p* < 0.001, women: *p* < 0.001), while the grip strength and JST-IC scores in the after driving cessation group were significantly lower (grip strength: *p* < 0.001, JST-IC: *p* < 0.001) compared with those in the active driving group. The proportion of those with apathy (*p* = 0.009), physical frailty (*p* = 0.001) and social frailty (*p =* 0.002) in the after driving cessation group was significantly higher compared with those in the active driving group. There was no difference in satisfaction (*p* = 0.266) or performance (*p* = 0.655) of meaningful activities between the two groups. Of the meaningful activity categories selected by the active driving and after driving cessation groups, the ratio that selected leisure was high in both groups (driving: 32.1%, driving cessation: 27.9%) (Figure 2). The work/education in the active driving group was significantly higher than that in the after driving cessation group (active driving: 5.9%, after driving cessation: 1.4 %, *p* < 0.05), while domestic life in the after driving cessation group was significantly higher compared with that in the active driving group (active driving: 16.3 %, after driving cessation: 27.9%; *p* < 0.01) (Figure 2). Regarding specific activities of domestic life, cooking meals (27.0%), collecting information (keeping up to date using newspapers and other news sources) (14.7%), and shopping (14.5%) were often chosen in the active driving group, and cooking meals (36.7%), shopping (11.7%), and laundry (11.7%) were often selected in the after driving cessation group (Table 2). In specific activities of work/education, remunerative employment (56.4%) and non-remunerative employment (38.5%) were often chosen in the active driving group, and remunerative employment (66.7%) was often selected in the after driving cessation group (Table 2).

To determine the effect of gender, we only analysed women. In women, the grip strength (*p* < 0.001) and JST-IC score (*p* < 0.001) were significantly lower, and the proportion of physical frailty (*p* = 0.001) and social frailty (*p* = 0.018) was significantly higher in the after driving cessation group compared with those in the active driving group (Table 3). Furthermore, in women, work/education was significantly higher in the active driving group compared with the after driving cessation group (active driving: 5.7%, after driving cessation: 1.1%; *p* < 0.05), and domestic life in the after driving cessation group was significantly higher than in the active driving group (active driving: 19.8%, after driving cessation: 29.4%; *p* < 0.05) (Figure 3). In contrast, for men only, “Leisure” was the most frequently selected activity for both the active driving group (35.6%, Age: 74.2 ± 6.0) and the after driving cessation group (36.8%, Age: 78.3 ± 6.9). Then, less frequent driving resulted in lower grip strength (*p* < 0.007), and more apathy (*p* = 0.001) and social frailty (*p* < 0.025) (Table 4). 

## 4. Discussion

As people get older, they may stop driving due to various reasons, such as deterioration in physical function. However, a driving interruption can be a turning point in their lives. Changes in the living environment due to interruptions in driving are affected by various factors such as physical and psychosomatic functions, personal factors, and environmental factors, and these complex interrelationships can affect daily life. In this study, we examined whether meaningful activities and psychosomatic function of community-dwelling older adults differ due to their driving status. We found that the active driving group valued work/education, while the after driving cessation group attached importance to domestic life. Moreover, the grip strength and JST-IC score were significantly lower in the after driving cessation group than in the active driving group. The proportion of apathy and physical and social frailty were significantly higher in the after driving cessation group than in the active driving group. In the active driving group, grip strength decreased, and social frailty increased as the frequency of driving decreased. As public transport in Tarumizu City is inadequate, it is challenging to live without a private car. Therefore, for older adults considering driving cessation in the future, it is necessary to support them considering their meaningful activities and intervene to maintain their physical activities.

Regarding the psychosomatic functions of the older adults who stopped driving, it was found that their grip strength and activity ability were considerably reduced. Furthermore, the incidences of apathy and physical and social frailty were high. Thus, we think that various daily life activities are restricted by interrupted driving, which may cause a decrease in muscle strength and the ability to be active as well as an increase in apathy in older adults.

Comparing the meaningful activities between both the groups, it was found that the active driving group attached high value to work/education, while the after driving cessation group attached importance to domestic life. As the active driving group was younger and had a higher proportion of men than the after driving cessation group, it was comprised of individuals who were the financial earners in their family. A previous study has reported that older people who retire from their full-time jobs have worse mental health (GDS-15) and Higher-Level Functional Capacity [45]. Therefore, the active driving group in this study may consider that continuing to work can be effective in terms of health maintenance and social participation. Psychosomatic health and daily and social activity could be maintained by continuing work. In contrast, it has been suggested that increasing the number of social participation activities and increasing participation in sports clubs and neighbourhood associations will prevent physical weakness in older adults [46]. For those who cannot continue their work or have already quit, it may be effective to work in the neighbourhood instead of at their jobs. Conversely, the ratio of women was higher in the after driving cessation group, and many women attached great value to domestic life. Previous studies reported that women were focused on domestic life higher than eight hours a week compared to men [47]. Further, women over the age of 65 also scored significantly higher than men in activities associated with domestic life [22]. There existed 80.0% of women in the after driving cessation group in this study. Domestic life and indoor activities are suggested to be meaningful activities after driving interruption for women. In contrast, women in the active driving group placed greater importance on work/education. Significant differences in activity before and after driving can have a negative impact on life after driving cessation. Therefore, interventions based on social cognitive theory with an emphasis on driving cessation plans and the involvement of friends and family as reported in previous studies may be effective for driving cessation [48].

Previous studies have reported that low mileage while driving [11] and the proportion of car accidents that occur after the age of 60 years are associated with physical frailty [12]. In this study, physical and social frailty were higher in the after driving cessation group than in the active driving group. Therefore, it is suggested that driving cessation may promote a decrease in social and physical activities [15]. Although previous studies have reported the accelerated decrease of cognitive function after driving interruption, there was no significant difference in Mini-Cog and cognitive frailty between the active driving and the after driving cessation groups in our study. This may be because the sample size of the after driving cessation group was small, and the duration after driving interruption was not evaluated in this study. In contrast, the after driving cessation group had increased apathy compared to the active driving group in this study. After driving cessation, they have fewer opportunities to go out, and meaningful outdoor activities are limited; hence, they may develop apathy. The authors acknowledge that the use of the apathy sub-scale of the GDS in this study is complicated by the fact that the items “dropping activities” and “staying home” are directly affected by driving cessation. Highly extensive and longitudinal studies need to be performed to investigate the effect of driving cessation on cognitive function and depressive status.

As drivers drive less frequently, their physical function declines and their social frailty increases. This means that even those who are currently driving are likely to drive less often and experience interruptions, increasing the risk of reduced activity, apathy, and physical frailty. Previous studies have reported that lower Short Physical Performance Battery (SPPB) scores were consistently with lower driving exposure and increased driving cessation [49,50]. However, physical function is a correctable risk factor and an increase in SPPB score can be achieved through fitness interventions [51]. Therefore, meaningful activity, including physical factors, may prevent the decline in physical function due to reduced or interrupted driving exposure. In addition, a decrease in driving frequency may be a sign of decreased activity. Therefore, it is necessary to pay attention to the frequency of operation and prevent the decrease in activity by meaningfully engaging them, in preparation for life after the operation is stopped.

Although driving cessation is associated with psychosomatic function in older citizens, pre-planning for driving cessation has been suggested to influence the quality of life of older persons [52]. Moreover, professionals need to participate in the process as evaluators of driving aptitude [53]. Therefore, occupational therapists should cooperate with driving license centres and communities to support older adults considering driving cessation. The results of this study indicate that we need to consider the meaningful activity of individuals and take measures to prevent apathy, muscle weakness, and reduction of activity through planning for driving cessation.

This study has several limitations. First, it is a cross-sectional study, and we cannot clarify whether our results are causes or results of driving cessation. Second, this study was carried out in one city, and we cannot deny the selection bias. Therefore, we need to perform high extensive and longitudinal studies to investigate the effect of driving cessation on meaningful activity and physical and social frailty. Third, the percentage of people after driving cessation in the final sample of the study was 8.2%, and the group sizes between active driving and after driving cessation were very different. This means that there is a higher scope for a wide variety of activities in the active driving group. However, in previous studies, the percentage of people who stopped driving was 9.0% [2], 1.4% [5], and 5.2% [6], and the after driving cessation group in this study was the same as or slightly higher. Fourth, we did not determine the factors and details of why older people stopped driving. The reasons to stop driving vary, and we need to analyse these reasons, including physical, social, and psychological frailty, in further studies.

## 5. Conclusions

In conclusion, we found that meaningful activity differed depending on the driving status; the active driving group valued work/education, while the after driving cessation group attached importance to domestic life. Moreover, in terms of physical and psychosomatic function, the after driving cessation group may be associated with weak muscle strength, apathy, and physical and social frailty. In addition, infrequent driving in the active driving group may give rise to an increasing number of older adults with social frailty. These results may be helpful in the pre-planning and support of driving cessation in older adults.

## Figures and Tables

**Figure 1 ijerph-18-13270-f001:**
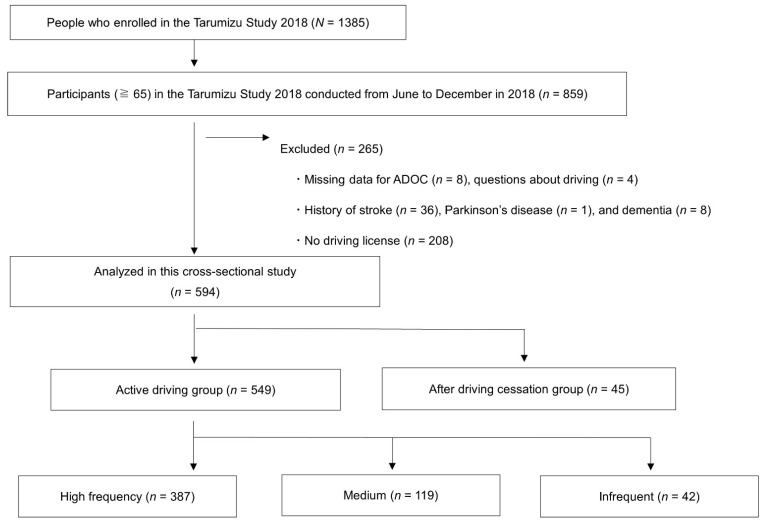
Flowchart of the present study.

**Figure 2 ijerph-18-13270-f002:**
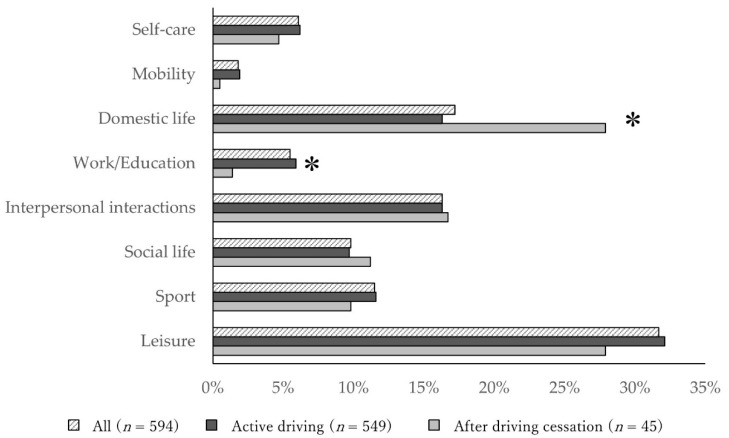
Comparison of the meaningful activities between the active driving group and the after driving cessation group in all participants. * *p* < 0.05, active driving group vs. after driving cessation group.

**Figure 3 ijerph-18-13270-f003:**
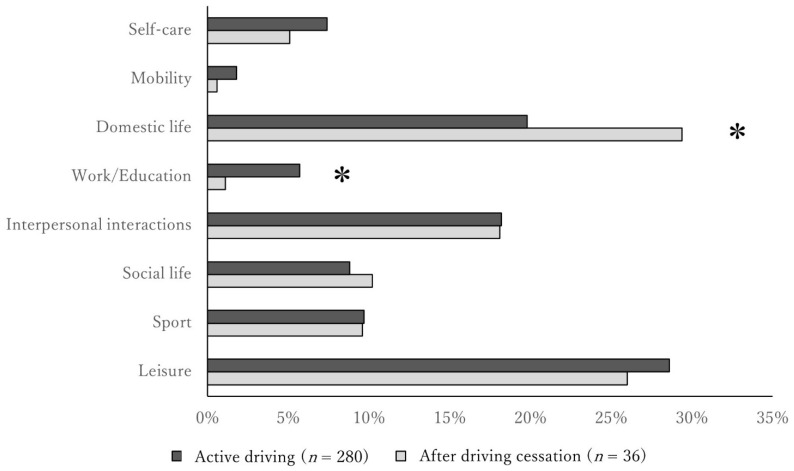
Comparison of meaningful activities between the active driving and the after driving cessation group in women. * *p* < 0.05, active driving group vs. after driving cessation group.

**Table 1 ijerph-18-13270-t001:** Comparison of characteristics between the active driving and the after driving cessation group in all the participants.

	Group		*p* Value
	Active Driving (n = 549)	After Driving Cessation (n = 45)	
Age (Years)	73.1 ± 5.6	77.1 ± 6.5	<0.001 a
Women, n (%)	280 (51.0)	36 (80.0)	<0.001 b
Education (Years)	11.5 ± 2.4	11.0 ± 1.8	0.131 a
Medication (Numbers)	3.56 ± 4.4	5.51 ± 4.3	0.005 a
BMI (kg/m^2^)	23.4 ± 3.2	23.3 ± 3.4	0.871 a
Grip strength (kg)	27.0 ± 7.0	21.8 ± 6.8	<0.001 a
JST-IC (Points)	12.1 ± 2.9	9.96 ± 2.7	<0.001 a
GDS (Points)	2.32 ± 2.4	2.95 ± 2.6	0.118 a
Depression, n (%)	80 (14.6)	10 (22.7)	0.147 b
Apathy, n (%)	115 (20.9)	17 (37.8)	0.009 b
Poor Cognition, n (%)	92 (16.8)	7 (15.6)	0.835 b
Living Alone, n (%)	119 (20.9)	18 (40.0)	0.003 b
Physical frailty, n (%)	7 (1.3)	5 (11.1)	0.001 b
Cognitive frailty, n (%)	47 (8.6)	7 (15.6)	0.102 b
Social frailty, n (%)	61 (11.1)	12 (26.7)	0.002 b
Satisfaction with Activity	4.0 (4.0–5.0)	4.0 (3.3–5.0)	0.266 c
Performance of Activity	8.0 (7.0–10.0)	8.0 (7.0–10.0)	0.655 c

Values are mean ± standard deviation (SD) or median (interquartile range); BMI, Body Mass Index; JST-IC, Japan Science and Technology Agency Index of Competence; GDS, Geriatric Depression Scale; a Student’s *t*-test, b Pearson’s χ^2^ test, c Mann–Whitney U-test.

**Table 2 ijerph-18-13270-t002:** Detail of meaningful activities related to domestic life and work/education in the active driving group and after driving cessation group.

	No.	Active Driving Group (n = 549)		After Driving Cessation Group (n = 45)
Domestic life	1	Cooking meals (27.0%)	1	Cooking meals (36.7%)
	2	Collecting information (14.7%)	2	Shopping (11.6%)
	3	Shopping (14.5%)		Laundry (11.6%)
	4	Cleaning (11.5%)	4	Cleaning (10.0%)
	5	Laundry (10.6%)	5	Collecting information (8.3%)
	6	Child Care (6.0%)	6	Management of property (6.7%)
	7	Assisting old people/patients (4.0%)	7	Making and repairing clothes (5.0%)
	8	Household maintenance (3.5%)		Child Care (5.0%)
	9	Management of property (3.0%)	9	Household maintenance (1.7%)
	10	Maintaining vehicles/appliances (2.5%)		Assisting old people/patients (1.7%)
	11	Makeup (1.4%)		Barbershop (1.7%)
	12	Making and repairing clothes (0.9%)		
	13	Writing a letter/document (0.2%)		
		Barbershop (0.2%)		
Work/Education	1	Remunerative employment (56.4%)	1	Remunerative employment (66.7%)
	2	Non-remunerative employment (38.5%)	2	Non-remunerative employment (33.3%)
	3	Informal education (4.5%)		
	4	School education (0.6%)		

The activities selected by the older adults are listed in descending order for the items that showed a significant difference in the comparison of meaningful activities.

**Table 3 ijerph-18-13270-t003:** Characteristics in the active driving group and the after driving cessation group in women.

	Group		*p* Value
	Active Driving (n = 280)	After Driving Cessation (n = 36)	
Age (Years)	72.1 ± 5.0	77.6 ± 6.6	<0.001 a
Education (Years)	11.3 ± 1.9	10.9 ± 1.7	0.243 a
Medication (Number)	3.18 ± 3.5	5.5 ± 4.2	<0.001 a
BMI (kg/m^2^)	23.1 ± 3.4	23.8 ± 3.5	0.273 a
Grip strength (kg)	22.3 ± 4.2	19.6 ± 4.1	<0.001 a
JST-IC (Points)	12.6 ± 2.6	10.2 ± 2.5	<0.001 a
GDS (Points)	2.4 ± 2.4	3.0 ± 2.5	0.181 a
Depression, n (%)	43 (15.4)	8 (22.9)	0.256 b
Apathy, n (%)	68 (24.3)	14 (38.9)	0.060 b
Poor Cognition, n (%)	35 (12.5)	4 (11.1)	0.534 b
Living Alone, n (%)	73 (26.1)	14 (38.9)	0.105 b
Physical frailty, n (%)	1 (0.4)	3 (8.3)	0.005 b
Cognitive frailty, n (%)	14 (5.0)	5 (13.9)	0.051 b
Social frailty, n (%)	24 (8.6)	8 (22.2)	0.018 b
Satisfaction with Activity	4.0 (4.0–5.0)	4.0 (3.3–5.0)	0.178 c
Performance of Activity	7.0 (7.0–10.0)	10.0 (8.0–10.0)	0.281 c

Values are mean ± standard deviation (SD) or median (interquartile range); BMI, Body Mass Index; JST-IC, Japan Science and Technology Agency Index of Competence; GDS, Geriatric Depression Scale; a Student’s *t*-test, b Pearson’s χ^2^ test, c Mann–Whitney U-test.

**Table 4 ijerph-18-13270-t004:** Characteristics of differences in the number of driving days per week.

	Group			*p* Value
	High-Frequency (n = 387)	Medium (n = 119)	**Infrequent** **(n = 42)**	
Age, mean ± SD (Years)	73.1 ± 5.5	73.2 ± 5.6	73.7 ± 6.5	0.745 a
Women, n (%)	185 (47.8)	70 (58.8)	25 (59.5)	0.057 b
Education, mean ± SD (Years)	11.5 ± 2.4	11.6 ± 2.2	11.6 ± 2.5	0.721 a
Medication ± SD (Number)	3.7 ± 4.8	3.2 ± 3.4	3.0 ± 3.4	0.319 a
BMI, mean ± SD (Kg/m^2^)	23.1 ± 3.4	23.8 ± 3.5	23.1 ± 4.0	0.273 a
Grip strength, mean ± SD (Kg)	27.5 ± 7.1	26.0 ± 7.1	25.0 ± 4.7	0.007 a
JST-IC, mean ± SD (Points)	12.1 ± 2.9	12.2 ± 2.8	11.6 ± 3.0	0.445 a
GDS, mean ± (Points)	2.2 ± 2.3	2.6 ± 2.7	2.8 ± 2.4	0.124 a
Depression, n (%)	48 (12.4)	22 (18.5)	10 (23.8)	0.055 b
Apathy, n (%)	66 (17.1)	33 (27.7)	16 (38.1)	0.001 b
Poor Cognition, n (%)	71 (18.3)	15 (12.6)	6 (14.3)	0.309 b
Living Alone, n (%)	81 (20.9)	27 (22.7)	7 (16.7)	0.711 b
Physical frailty, n (%)	4 (1.0)	2 (1.7)	1 (2.4)	0.690 b
Cognitive frailty, n (%)	30 (7.8)	16 (34.0)	1 (2.4)	0.050 b
Social frailty, n (%)	35 (9.0)	17 (14.3)	9 (21.4)	0.025 b
Satisfaction with Activity, Median (IQR)	4.0 (4.0–5.0)	4.0 (4.0–5.0)	4.0 (3.0–5.0)	0.289 c
Performance of Activity, Median (IQR)	8.0 (7.0–10.0)	8.0 (7.0–10.0)	9.0 (7.0–10.0)	0.642 c

SD, standard deviation; BMI, Body Mass Index; JST-IC, Japan Science and Technology Agency Index of Competence; GDS, Geriatric Depression Scale; IQR, interquartile range; a one-way ANOVA, b Pearson’s χ^2^ test, c Kruskal–Wallis test. High-frequency group (6–7 days a week), Medium group (3–5 days a week), and Infrequent group (2 days or less a week).

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
