# Peer review of "Meaningful Activities and Psychosomatic Functions in Japanese Older Adults after Driving Cessation"

_ijerph, 2021, doi:10.3390/ijerph182413270_

Round 1

Reviewer 1 Report

Summary

This group has collected extensive information from a number of older adults living in the Tarumizu region of Japan. They compared physical, cognitive and mental health dimensions and preferred activities in a large group of active drivers to a small group of adults no longer driving. Interesting data is provided for the female sub-group.

Major issues

Driving cessation is an “act” that happens at one point in a person’s life. If this was a prospective trial – there would be a point where “driving cessation” happens. But in this trial since there is no “intervention” to stop driving, and since there is no idea for how long the older adults have ceased driving – it would be better to refer to the two groups as “active driving” and “no longer driving.” The title should read: “… in Japanese Older Adults After Driving Cessation” or better yet “… in Japanese Older Adults No Longer Driving.”

While the last section in the Discussion rightly states that from a cross-sectional study it is not possible to determine “cause” and “effect” – for instance does a weakness in grip strength lead to unsafe driving and hence cessation, or does the decision to stop driving lead to decrease in physical activities and ultimately decreasing grip strength. If this study had determined how long the driving had been stopped, a sub-group analysis could have been performed. There are a number of statements in the text that imply causality: Abstract line 32 “resulted in decreased,” line 281 “apathy increased after driving interruption,” line 315 “social frailty increased.” Please modify these to show a relationship without implying causality.

The methods section does not define high, medium, and infrequent driving. 

It is unclear why the sub-analysis of the male group was not provided – especially since men would presumably not have valued domestic activities as much. I realize the number would be small (n=9) and maybe p values would not be significant, but at least a description would be interesting.

I have a concern about using the “apathy” sub-scale of the GDS in this case – since anyone who is no longer driving is “by definition” staying at home more and doing fewer outside activities. If you decide to keep this – please explain how to interpret this.

I am confused about line 146 where a decline in speed (presumably walking?) and weight (not sure if weight loss or lifting weights?) somehow is linked to cognitive frailty. This requires more explanation for those of us not familiar with the tools. Similarly in lines 149 and 150 it is not clear how “going out less frequently” and “talking to someone every day” are pointing in the same direction regarding social frailty.

The Tables need to be checked – there are some formatting issues e.g., “Age” in Tables 1 and 3, and there are a number of errors in Table 4: “Infrequent” column for “BMI,” “Apathy” and “Satisfaction with Activity” (missing parenthesis) – and the p value for “Satisfaction with Activity.”

Minor issues

Line 47: “having declining independence in Activities of Daily Living”

Line 57: provide more information about the low population density: either the actual number or at least suggest how large the area is where these 14K people live.

Line 67: please clarify this sentence

Line 76: replace “driving situation” with “driving status” throughout the paper. Also replace “whether to drive” with “driving or not driving.”

Line 78: replace “outages” with “outings”

Line 109: I don’t understand what is meant with “was listened to using an iPad application.” Were the questions recorded on the iPad – then how were the answers recorded?

Line 186: I don’t understand what “collecting information” means.

Line 190: please explain why you think “remunerative employment” was often selected by the “no longer driving” group.

Line 205: maybe replace “exclude” with “determine”

Line 214:” grip strength was decreased and social frailty was increased.”

Line 252: “ability of activity as well as…”

Line 254: “In terms of comparing”

Line 278: “difference?”

Line 283: I understand what is meant, but theoretically when driving ceases, there could be more walking to do activities like groceries. Presumably walking to the grocery sore would be “a meaningful outdoor activity.”

Line 303: “selection bias”

Reviewer 2 Report

This paper has three qualities. It is very concise and structured in its presentation. Its study objective is of great interest. Analyzing the impact of driving cessation on health and lifestyle is of great interest. The analyses are adequate to address the objectives formulated.    

The fundamental limitation of the work is, as the authors themselves point out, that it does not allow us to know the direction of the relationships between the variables. However, this is a limitation shared with other published works.  

On the other hand, I consider that the work could be substantially improved if the following issues were clarified.  

Lines 91-93 mention cases excluded due to medical diagnoses, why these and not other diagnoses? 

line 99 (and Line 96-97) Clarify the description of the participants who were not driving (n=16).   99 These participants could also be part of the driving group described in Lines 96-97:  

I suggest clarifying the description of driving group on lines 96-97 or clarifying description of participants mentioned on line 99. 

Consultation on some references (I note some examples) 

I have not been able to complete the reference [27] Is it possible that there is an error? 

[27 Yesavage, J.A. Geriatric depression scale. Psychopharmacol Bull 1988, 24, 709-711].  

Reference [26] is cited in lines 111-112. to illustrate validity of satisfaction 

in the evaluation. This study does not address the validity of satisfaction. Is it possible that there is some error?  

110 ...and the validity of satisfaction 

111 in the evaluation has been reported [26]. 

Regarding the description of the instruments: 

Regarding the measure of meaningful activities (ADOC):  

-It is appropriate to justify the relevance of applying the ADOC to a non-institutionalized sample. Since it is an instrument aimed at Community-dwelling persons. I know that there are previous works that use it, but it is convenient to justify this particularity. 

-On the other hand, the ADOC is an instrument that makes it possible to detect significant activities in collaboration with the therapist and with a view to rehabilitation. However, the participants in the study were not described as rehabilitation patients, nor was the existence of therapists.  

Regarding the therapists, it should be clarified whether the therapists are the well-trained staff members investigated the meaningful activities. mentioned in lines 120-121. Clarify in some way how their degree of training was assessed. 

Satisfaction Measure  

This variable should be described. It is not clear whether it is a measure of overall satisfaction with life or with performance or with the resulting list of activities mentioned in lines 120-121.  

Line 235 Revise this statement: 

Driving cessation due to old age is a turning point in life. Since the driving situation 235 

This statement suggests that people stop driving because of old age. This does not seem to be the case. In the study itself they have only been able to identify 45 vs. more than 500 who maintain the activity. It is suggested to use an expression that does not attribute a causal role to age but, perhaps, to other contingent factors. 

Line 316 erratum "vulunerabilities". 

Reviewer 3 Report

Introduction: Author's state that driving cessation leads to poor health, but the relation between health, functioning and driving cessation is more complex than that and vice versa also holds true. The introduction needs to reflect these complex interactions.

This sentence starting on line 57 does not justify the sentence that follow on line 59: Tarumizu City, with a population of 14,379 and a low population density, is one of the least populated areas of Japan, and the population aging rate there is extremely high. The population density and ageing does not justify why driving is necessary.

The sentence starting on p 61 describes necessary activities rather than meaningful activities. Meaningful activities entail subjective evaluation, and that is not clear from the introduction.

The authors state on line 69-70: However, there are no reports on meaningful activities for older adults who have stopped driving or who drive infrequently. However, this statement is not true and there is literature to find that should be referred to.

Methods: How participants were sampled and recruited is not clear enough. In the participants section it seems that participants were interviewed before inclusion and informed consent. The first sentence on line 101 needs to be checked language-wise.

The data collection is not sufficiently described. For example it does not make sense that ADOC was listened to using an iPad application. The reader needs to understand better how the instrument is designed and data collection should be conducted using that instrument. To say that ADOC is structured (line 115) is quite an empty information, what do the authors intend to say? Moreover, is not how the staff members were trained, what does it mean to be well-trained. Did the participants self-report or were they observed using ADOC?

Moreover the final sample is too skewed (the groups is of very different size) to support reliable findings, and the authors does not recognise that 

The following sentence does not make sense: Frailty encompassed three aspects of the participants (line 136).

Leisure activities is not sufficiently defined.

Table 2 says to report significance, but information on significance is missing.

Discussion:

The claim on line 235 Since the driving situation 235 influences the living environment, is not supported in the text, and that needs to be done.

Overall the discussion is too limited and includes far to little discussion based on other references.

It partly includes too much of speculation for example on line 260: Therefore, the driving group in this study may consider that continuing work can be effective in terms of health maintenance and social participation. 

On line 267 the text reads: Domestic life and indoor activities are suggested to be meaningful activity after driving interruption for women. It deserves to be problematised that meaning change after driving cessation. 

Round 2

Reviewer 1 Report

I appreciate that the authors have made significant improvements to the paper. There are only a couple of minor issues that should be changed before publication:

I agree with the authors that use of the “apathy” sub-scale of the GDS is well established. However, I still feel strongly that its use in this case is complicated by the facts that the activities are impacted de facto with driving cessation.  Maybe a compromise would be to include a statement in the discussion around line 322, add a sentence “The authors acknowledge that the use of the apathy sub-scale of the GDS in this study is complicated by the fact that the items ‘dropping activities’ and ‘staying home’ are directly affected by driving cessation.”  

Line 162: given that line 158 now defines that frailty has a number of dimensions, can we say for line 162 “Combined physical and cognitive frailty is defined as ….” and use this throughout the paper? Cognitive frailty has its own definition with only cognitive components.

Please be more explicit in line 169: if m/s is walking speed and if pounds is lifting/pressing weights with upper/lower limbs.  

Line 210: add: collecting information (keeping up to date using newspapers and other news sources).

Please check if lines 238 and 239 are in the correct location – lines 240 to 242 now look like they refer to men only.

Reviewer 3 Report

Comments added in the file

Author Response

This manuscript is a resubmission of an earlier submission. The following is a list of the peer review reports and author responses from that submission.